# An Integrated Fusion Engine for Early Threat Detection Demonstrated in Public-Space Trials

**DOI:** 10.3390/s23010440

**Published:** 2022-12-31

**Authors:** Henri Bouma, Maria Luisa Villani, Arthur van Rooijen, Pauli Räsänen, Johannes Peltola, Sirra Toivonen, Antonio De Nicola, Massimiliano Guarneri, Cristiano Stifini, Luigi De Dominicis

**Affiliations:** 1TNO-Netherlands Organisation for Applied Scientific Research, 2597 The Hague, The Netherlands; 2ENEA-Italian National Agency for New Technologies, Energy and Sustainable Economic Development, 00123 Rome, Italy; 3VTT-Technical Research Centre of Finland, 33101 Tampere, Finland; 4ATAC-Azienda per la Mobilità di Roma Capitale S.p.A, 00176 Rome, Italy

**Keywords:** counter terrorism, surveillance, threat detection, forensics, re-identification

## Abstract

Counter terrorism is a huge challenge for public spaces. Therefore, it is essential to support early detection of threats, such as weapons or explosives. An integrated fusion engine was developed for the management of a plurality of sensors to detect threats without disrupting the flow of commuters. The system improves security of soft targets (such as airports, undergrounds and railway stations) by providing security operators with real-time information of the threat combined with image and position data of each person passing the monitored area. This paper describes the results of the fusion engine in a public-space trial in a metro station in Rome. The system consists of 2D-video tracking, person re-identification, 3D-video tracking, and command and control (C&C) formulating two co-existing data pipelines: one for visualization on smart glasses and another for hand-over to another sensor. Over multiple days, 586 commuters participated in the trial. The results of the trial show overall accuracy scores of 97.4% and 97.6% for the visualization and hand-over pipelines, respectively, and each component reached high accuracy values (2D Video = 98.0%, Re-identification = 100.0%, 3D Video = 99.7% and C&C = 99.5%).

## 1. Introduction

Terrorism is an unpredictable international security challenge for crowded soft targets, such as airports, train or metro stations, and shopping centers [1,2]. In these sites, examples of possible threats are people carrying firearms or explosives. However, most of the existing systems capable to detect threats [3] require time consuming one-by-one assessments, which leads to long queues that are not compatible with the daily life of citizens and commuters.

DEXTER is a flagship initiative of the NATO Science for Peace and Security (SPS) Program [4]. In the DEXTER program, a system was developed that can remotely and in real time identify the carriers of explosives and firearms in public crowded venues. The DEXTER system consists of three main components: MIC, EXTRAS and INSTEAD. MIC is a sensor for the detection of firearms and bulk explosives and EXTRAS is a sensor for the detection of explosive traces. INSTEAD is an integrated fusion engine for the management of a plurality of sensors to detect threats without disrupting the flow of commuters. INSTEAD consists of 2D-video tracking, person re-identification, 3D-video tracking, fusion and command and control.

The problem of developing an integrated surveillance system is considered relevant by the research community [5], but most of the existing approaches [6,7] rely on simulation-based methods. To our knowledge, there is a lack of scientific publications where the results of real case studies are presented and discussed. The DEXTER system was tested in the metro station Anagnina in Rome in May 2022. Anagnina is a central metro hub as it collects commuters from a wide region (about 28.000 per day) to attend their work in the center of Rome.

This paper focuses on the INSTEAD system and describes the results of its usage in the public-space trial at the Anagnina metro station. Coping with the complexity of managing data from different sources and the relevance of such type of data having an impact on security of people pose several data challenges [8], such as volume, variety, velocity, availability, privacy, and scalability. The research question addressed in the paper is to what extent the INSTEAD system performs well in a real environment, such as a crowded metro station. To this purpose, we evaluated the accuracy of the different INSTEAD components and we discussed the results in detail to set a benchmark for future studies.

This paper contains the following key contributions. A novel fusion engine is demonstrated and tested in a large-scale experimentation with real persons in a metro station and the results are described in this paper. Another contribution is that the fusion of information from distributed sensors is enabled by a real-time video-based matching of people at different locations, which does not even require overlapping fields-of-view. A smart retraining strategy is used to obtain high-quality matches with privacy-enhancing re-identification technology in the metro environment. The last contribution is the compensation for location bias and latency to support an accurate and timely hand-over between sensors.

The rest of the paper is organized as it follows. Section 2 gives an overview of related work. Section 3 describes the fusion engine. Section 4 describes the experimental setup. Section 5 presents the results. Finally, Section 6 summarizes the conclusions and suggestions for future work.

## 2. Related Work

Threats can occur in the physical world and in cyberspace. Others already created an excellent overview of insider threats, which are malicious cyber threats from people within the own organization [9]. This paper focusses on the early detection of physical threats in public space to protect soft targets and the fusion of multiple sensors.

Data fusion is defined as the integration of information from multiple sources to produce specific and comprehensive unified data about an entity [10]. Fusion of data from different sensors to detect explosive devices has been already addressed by literature. Deiana and Hanckmann [11] proposed a framework for data fusion, which includes sensors based on different technologies, such as millimeter wave imaging, radar technology, THz technology and infrared sensors. Similarly, Knox et al. [12] proposed an approach that uses precise location information and includes ground penetrating radar and electromagnetic induction sensors. Volpetti et al. [13] developed a prototype aimed at CBRNe (Chemical, Biological, Radiological, Nuclear and explosives) threat detection and monitoring, which consists of beacons embedding innovative smart sensors and a command centre aimed at data fusion and real-time visualization of geo-referenced alarms. Sensor fusion aimed at explosive hazard detection was also addressed by Pinar et al. [14]. In detail, they focused on reducing the number of false alarms by adopting a multiple kernel learning support vector machine (SVM) classifier and tested the approach by means of governmental data. Finally, Frigui et al. [15] discussed, tested, and compared seven fusion algorithms for detecting anti-tank landmines. These are: Bayesian, distance-based, Dempster-Shafer, Borda count, decision template, Choquet integral, and context-dependent fusion. With respect to the above-presented works, the data fusion method presented in this paper is generic as it refers to real-time correlation of sensor results with video messages on the targeted commuter. Furthermore, we tested our approach in a large-scale experimentation with real persons in a metro station. Finally, we present the final results that could be used as benchmark for future studies.

Others have shown that fusion can be performed at a location with overlapping fields-of-view [11,12]. In this paper, we use video-based person re-identification technology to support fusion of sensor information from different locations without overlapping fields-of-view. One approach to perform a forensic search for people on digital video material is with face recognition technology [16]. However, the resolution in many CCTV cameras is often not sufficient for reliable face recognition, people can look in another direction and it has a high impact on the privacy of citizens. Re-identification technology can be applied to lower-resolution images because it relies on cues related to clothing and it is commonly intended to support a forensic search in an interactive way. In this paper, we use the re-identification technology for the fusion of sensors in a fully automatic way. The privacy of citizens is enhanced by anonymizing faces and recent work showed that it hardly deteriorates the quality of re-identification technology [17]. Others already showed that re-identification can reach a high Rank-1 accuracy on public datasets [18], but the performance deteriorates when the pre-trained model is applied in another environment. Recent progress shows that the precision and recall values can be higher than 99% in a new environment when applying a smart automatic retraining strategy [19]. In this paper, a large-scale evaluation is performed in the metro environment to assess the performance of the full automatic matching of people at different locations without overlapping fields-of-view to support the fusion of sensors.

3D sensors provide an estimation of distance, and using this additional information improves human tracking accuracy compared to normal 2D cameras [20]. Most of the published research is focused on accurate motion tracking [21,22] and is targeted on improving relative accuracy in modelling movement of limbs relative to other body parts. Our solution is optimized for absolute accuracy in a common coordinate system that enables robust sensor fusion for selecting the correct person in data of multiple sensor types which is essential to perform a successful handover. In addition, we have developed a motion predictor that improves fusion accuracy by compensating for latency in communications and mechanical adjustment of Pan/Tilt sensors. In this paper, we assess the quality of the hand-over between sensors.

## 3. Fusion Engine

This section describes the INSTEAD system. First, the architecture is described (Section 3.1) and then each of the main components (Section 3.2, Section 3.3 and Section 3.4).

### 3.1. DEXTER and the INSTEAD Architecture

The DEXTER system encompasses two sensors, MIC and EXTRAS, that deliver quantitative data to help identifying a perpetrator of a terroristic attack with explosives or weapons before the person can finalize the attack. In a deployment of DEXTER, these two detectors are positioned at the start and end of a pre-defined corridor, so that any MIC positive commuter can be readily identified and tracked by two video systems until he/she reaches the EXTRAS sensor that will complete the detection result. Thus, threat commuters are identified very close to the monitored area. The alarm management system—which delivers information to the security guards—is triggered in real time.

The role of the INSTEAD system is to integrate sensors (MIC, EXTRAS) and trigger the alarm management system for informing security guards in a timely manner. The INSTEAD architecture is shown in Figure 1, where Sensor 1 is MIC and Sensor 2 is EXTRAS.

INSTEAD is a decentralized system where the various components communicate by exchanging messages according to a publish-subscribe protocol [23]. The 2D video system is responsible for commuter identification in the area near MIC, tracking of the commuter, and of later re-identification in the area near EXTRAS (Section 3.2); the 3D video system for subsequent tracking and location prediction of the commuter closer to EXTRAS (Section 3.3); and Command & Control (C&C) for fusion and communication (Section 3.4). The components of INSTEAD are described in more details in the following subsections.

### 3.2. 2D Video Tracking and Re-Identification

Person re-identification is a technology that can match people based on similarity in different cameras (Figure 2). On one hand, the technology can be used interactively by police officers to find a suspect in the neighborhood. On the other hand, it can be used to automatically track people from one camara to another, even if the cameras are not overlapping. Recent progress shows that people can be matched even in a privacy-preserving way with anonymized faces [17] and the precision and recall values can be higher than 99% [19].

The architecture of the 2D video pipeline and its components is shown in Figure 3. These components are important to understand the results and error causes that are presented in Section 5.
-*2D cameras*: Multiple cameras generate a continuous video stream.-*Person detection and tracking*: The video stream is processed in real-time, and all persons are detected. Each person is tracked within a camera stream to obtain a more compact representation of information for a person. Faces are anonymized by applying a median filter.-*Re-identification (Re-ID) algorithm*: The Re-ID algorithm matches an image of a person in one camera with similar images in another camera.-*Publish messages*: The creation of JSON messages [24] includes the assignment of an identification (ID) number for each person.-*C&C communication*. The communication between 2D Video and C&C is facilitated by MQTT [25].

More details about these 2D Video components can be found in another paper [19].

Two assumptions were made in the design of the system for the controlled sessions of the big-city trial. The first assumption is that people are passing the sensors one-by-one with a time of approximately 2.0 s between the people to avoid occlusions. The second assumption is that the same person with the same clothing is not reappearing in the same environment for 4 min, to allow greedy matching and to avoid forced incorrect matches.

### 3.3. 3D Video Tracking

A 3D sensor consists of a 2D video stream and depth measurements. Two different 3D sensors are used: the first is used to make predictions (Section 3.3.1) and the second is used to control the EXTRAS sensor (Section 3.3.2).

#### 3.3.1. ZED2 Stereo Camera for Predictions

The system deployed in 3D Video tracking comprises of a commercial StereoLabs ZED2 stereo camera [26] connected to an edge computing unit based on NVIDIA Xavier computer (Figure 4). The system includes a simple software application built on top of ZED software development kit (SDK) in the Python programming language that detects humans as monitored objects and provides an estimation about the 3D center position of the head of observed humans. The head has been chosen as the tracking target because it is the most visible and non-occluded body part in crowded scenes.

The application tracks all the observed human objects in an unobtrusive manner while preserving privacy; tracks are stored in numerical 3D coordinates with anonymized object identifiers only. When the 2D video tracking system triggers an event about the monitored person, the 3D video system completes a handover from 2D-to-3D coordinate space. The handover deploys an algorithm in which both temporal and spatial information is matched between coordinates in the 2D trigger event and all human objects in 3D space. When there is a handover match, the 3D Video system starts computing location predictions for the monitored object (Figure 5).

Location predictions utilize a machine learning model based on a recurrent neural network (RNN) which is capable to cover straight, curved and wobbling tracks. The model uses coordinates of the 3D Video as input and estimates the location of the tracked person to a forward point in time. A forward offset of a second was found the most applicable in the experimental setup.

The location predictions are presented in world coordinates. The mapping from the 3D video space to the world coordinates is calculated using a computationally efficient affine transformation method. A conversion matrix was used to minimize bias [27].

In addition to the benefits of future location prediction, the prediction procedure also compensates the latency of communication and processing. To optimize computational performance, the model runs with NVIDIA TensorRT library [28] on an NVIDIA Jetson-Xavier.

The communication interface to 2D Video and C&C uses the MQTT protocol [25] and the communication to EXTRAS (for the predications) uses the UDP protocol to minimize communications latency.

#### 3.3.2. 3DSentinel for Monitoring and Control

An open hardware/software solution—called 3DSentinel—was developed for monitoring and for control of external devices (e.g., aiming the laser of EXTRAS). The adopted hardware solution is based on a Stereolab Zed2 Camera connected to a NVIDIA Jetson-Xavier Edge-PC, all assembled inside a 3D printed shell (Figure 6).

The software module is composed by a user interface (UI) (Figure 7), developed in PyQt5, and several algorithms for people tracking, object detection and classification. The software has a client/server structure, and the UI modifies its aspect based on the role it has on the computer where it is running. The communication with the external world is ensured by several protocols, including TCP and UDP. The video-streaming captured by the camera is ensured by packetizing the image size in several arrays with a 16 bits maximum length. The 3DSentinel software encapsulates several calibration methods for synchronizing with external devices and georeferencing the provided data. The common strategy used for referencing the data is based on the rectification of the depth information, estimated by the stereo camera, and by computing a transformation matrix.

Another interesting feature of the 3DSentinel is the possibility to save data locally, which can be retrieved afterwards for further analysis or simulation. When connected to external devices, especially robotic ones, it is possible to simulate back all the physical process which brought to a particular result.

Depth estimation, objects classification and people tracking are actually processed by the internal Stereolab Neural Network Engines, while the future releases will include also the possibility to use open-source architectures, like YOLO4 [29], and commonly used machine learning platforms, like Tensorflow [30] and PyTorch [31].

### 3.4. Command and Control

The Command and Control (C&C) is a distributed software component with the roles of orchestrating the functions of the technological components for threat detection and supporting the security agents in the alarm management that may follow. To this aim, the C&C allows for scene reconstruction for each commuter [32] and supports interoperability of heterogeneous IoT systems [33]. The C&C provides the following functions (as indicated in Figure 1).

Message exchange is coordinated by means of a Message broker implementing the MQTT standard protocol. The HiveMQ technology [34] has been adopted to this purpose.

Real-time data fusion combines sensor-data with video-based detection messages as soon as they are transmitted, and the components of the subsequent data flow are notified about results related to suspect people. Every detected commuter is assigned an Instead ID to correlate all the correct messages received by the video systems and threat detectors referred to him/her. A data-stream integrator server, such as WSO2 data-stream integrator [35], hosts the Data fusion and triggering component.

Alarm management is performed by the Security Client to compose threat information directed to the security interfaces, including the smart glasses (EPSON Moverio BT-40) of the security guards, and to receive confirmation messages from them.

Monitoring of the system is performed with Grafana dashboards [36] to visualize events of interest of system performance at run time captured by Prometheus temporal database server [37]. Furthermore, persistent storage of all the messages received by the C&C is provided for ex-post performance analysis of the system.

The UML state-chart in Figure 8 specifies the data fusion method implemented by the C&C in a normal functioning of the overall system. In particular, the model specifies the life-cycle of a single commuter detected by the system, i.e., the behavior of a “sensed” commuter. The states of a commuter and the state transition rules that are relevant for the integrated system are highlighted in the diagram. The state transitions are mostly enabled by events concerning the receival of messages from the sensors and the video system, and they are conditioned by the values in the detection results.

The following is a high-level description of the system behavior.

MIC and the 2D Video operate independently by observing a common area. MIC may generate a weapon detection with some level of confidence and send one or more messages to the C&C referring to that detection with improved results (state: *MICDetected*). In particular, each message specifies a time estimation when the intercepted object would be at a pre-specified line of the observed area. The 2D Video camera labels the person when the person passes a trigger line and notifies the C&C (state: *Labelled*), which then assigns an identifier to the commuter (state: *Commuter identified*). The C&C eventually fuses the 2D Video message with a selected MIC message that provides the closest detection time in a predefined (small) time range, if any such message exists. In this case, the commuter is considered MIC-positive and the C&C generates immediately an alarm for the SG with the image of the person.

After the commuter is re-identified by the 2D Video before EXTRAS, in case he/she is also MIC positive, the commuter is considered of high priority for EXTRAS (the state is *HighPriority Commuter*, otherwise the state is *Commuter NOT_Positive*). The C&C triggers the handover from 2D Video to 3D Video (state: *3DVideoDetection*) communicating the position of the commuter and, in case, the priority status.

EXTRAS eventually transmits a detection result to the C&C (state: *EXTRASDetected*), either positive or negative. The commuter is registered as a threat (state: *Threat*) if it is positive to either MIC or EXTRAS or both and an alarm is sent to the smart glasses if it was not sent before. The 2D Video sends a message to the C&C when the person exits the field of view of the second camera. In case the C&C has not received an EXTRAS result, a timeout establishes the end of the overall detection process. The data fusion method, incorporating also exceptional scenarios, has been detailed and implemented by concurrent Siddhi applications running in WSO2 data-stream integrator server [35].

## 4. Experimental Setup

### 4.1. Use Case and Scenario

The challenges posed by DEXTER are very ambitious and it was possible to reach the goals. DEXTER aims for remote and real-time detection of firearms and explosives carried by pedestrians, without requiring random checks on moving passengers or checkpoints in crowded venues and mass transit scenarios. Furthermore, DEXTER aims for integration of multiple technologies into an infrastructure capable of incorporating new and upgraded detection systems in the future, to have the potential of keeping up with evolving threats.

To test the DEXTER system (consisting of MIC, EXTRAS and INSTEAD), groups of volunteers walked through the trail area with varying attributes. A fraction of the volunteers carried weapons or bulk explosives concealed by clothing to test MIC and a fraction of the volunteers carried explosive traces to test EXTRAS. During the trial, challenging scenarios were inserted to test the limits of the system. For example, by letting commuters come close together to introduce occlusions, overtaking to verify re-identification, change of appearance of the commuters during the walk, varying number of commuters and commuters dressed alike at the same time in the corridor. These stress-tests were useful to determine the operational limits of the systems individually and combined.

### 4.2. Location

The trial was performed at the metro station Anagnina in Rome in May 2022 because this is a very busy hub. The INSTEAD system was already installed and tested as a separate component before the actual trial period. One and a half week was needed to integrate with MIC and EXTRAS, to move the technologies developed in laboratories to a real environment, face the external conditions (from cold to hot, humidity, ashes, network failures), and to prepare them for testing. The trial was a complex operation involving many people with their own expertise from different organizations to optimize the involved technologies. Trials and their preparation were attended by various teams for a total of about 45 people from 8 different institutions. For reasons related to safety (lasers, microwave), security (theft), legal (GDPR) and ethics (privacy), the area was delimed and only participants could enter the area. The people that participated during the trial are volunteers that gave their consent. A map of the environment is shown in Figure 9.

A common space-time reference model for all DEXTER components is required to be able to correctly correlate and interpret the exchanged data. As time reference, the Universal Time Coordinates (UTC) was chosen, and it is provided by a Network Time Protocol (NTP) server allowing computers clocks synchronization of the DEXTER private network. As space reference, world coordinates in meters were used relative to a pre-defined reference point in the environment. Three regions and detection lines within the corridor were defined to enable handover between components:*Init region*: A line to enable a hand-over between MIC and INSTEAD.*Re-ID region*: A line to enable a hand-over between 2D Video and 3D Video.*Exit region*: A line to enable a hand-over between from INSTEAD to EXTRAS.

### 4.3. User Interfaces

INSTEAD fuses the results of MIC and EXTRAS detections with images of the video and position data of each person passing in the monitored area to provide security operators with real time information of the person. Security guards wearing smart glasses (Figure 10 (left)) are notified of a threat by receiving the image of the commuter over imposed on their real view when she/he is still near the sensors, within the corridor in case of MIC positive result or at the exit in case of EXTRAS positive result only. An example of image of a commuter displayed on the security glasses is shown in Figure 10 (right). By pressing on the green and red flags the guard may confirm whether the displayed commuter has been recognized in the real environment.

For demonstration and performance assessment, additional user interfaces were provided to monitor and display detailed results (Figure 11 and Figure 12).

### 4.4. Schedule with Runs

The DEXTER system was tested for 9 days in 287 runs and 586 commuters (Table 1). On average there were more than 30 runs per day. Each run consisted of a small group of up to 4 people. The group walks from the MIC sensor towards the EXTRAS sensor. The commuters passed the MIC sensor one-by-one with a time interval of at least 2.0 s. Between the MIC and the EXTRAS sensor, commuters were allowed to overtake each other to challenge the Re-ID system. Near the EXTRAS sensor, people stopped a short time to facilitate the scanning. The time between one run and the next run was at least 4.0 min, to avoid reappearance of the same person within a pre-defined time interval.

## 5. Results

The internal performance of INSTEAD is assessed in two pipelines (Figure 13). The first is called the visualization pipeline, which directly supports the submission of images towards the smart glasses in case of threat suspect from the first (MIC) sensor. This pipeline includes the 2D Video and C&C components of INSTEAD. The second is the hand-over pipeline, which supports a handover from INSTEAD to EXTRAS. This pipeline consists of the 2D Video, C&C and 3D Video components.

The performance of the components inside INSTEAD (i.e., the two pipelines) are described in Section 5.1 and Section 5.2. The performance of communication outside INSTEAD (i.e., communication with MIC and EXTRAS) is described in Section 5.3. The computational and communication costs are described in Section 5.4.

### 5.1. Results of the Visualization Pipeline

The results of the visualization pipeline are shown in Table 2. The table shows an overall accuracy and a component accuracy. The overall accuracy is the performance of a cascade of all components relative to the total number of commuters (including errors of previous components). The component accuracy is the performance of one component relative to its own input (excluding previous components). The overall accuracy of the complete visualization pipeline is 97.4%, since there were 15 errors on 586 commuters. 2D Video has a component accuracy of 98.0% (12 errors) and C&C has a component accuracy of 99.5% (3 errors).

A more detailed analysis of the 2D Video errors is shown in Table 3. There are 12 errors, and the most dominant cause was due to occlusion near the MIC sensor.

The 2D Video errors can be solved in the following way:*Detection*: A separation between a trigger-line mode and a group mode was implemented [19]. In the trigger-line mode, a time interval of 2.0 s is required between the volunteers when passing the MIC trigger line, as defined in the planned scenario. The group mode appears to be robust for occlusions in crowded environments.*Reappear*: If the same person reappears in the last camera within 4 min, the results are unpredictable because it will return the best match in this time frame with a greedy algorithm. The system may return the second appearance in the first camera or the first appearance.*Re-ID alg.*: No errors due to the re-identification algorithm itself.*Publish*: Implementation of person ID assignment was re-implemented to generate unique ID’s and unique image URLs after restart of the system. Initially, the 2D Video reused old IDs after a restart. Later, this bug was fixed. IDs were made unique by adding an offset based on the time of restart.*C&C:* C&C could become stricter in the time-out to minimize dependencies of other modules.

So, in summary, the performance of 2D Video was good (98%) and the performance of the re-identification algorithm was perfect (100%). Most technical errors were fixed during the trials. The new group mode [19] makes the system more flexible to handle occlusions in crowded environments.

The C&C component accuracy is 99.5%. The acceptance criterium used for C&C evaluation is that for every 2D Video messages (i.e., Init, Re-ID and Exit) a correlating ID number must exist and only one. There is evidence in the data of 3 failures of the C&C in Day 7, but the root causes have not been clearly identified. In two of the three cases, the C&C processing stopped after MIC-2D Video data fusion but the complete output with the overall data of the commuters was not achieved.

### 5.2. Results of the Hand-Over Pipeline

The results of the hand-over pipeline are shown in Table 4. The table shows an overall accuracy of the hand-over pipeline of 97.6%. 2D Video has a component accuracy of 98.5%, as only 9 failures out of the 12 described in Table 3 hindered the hand-over. C&C has a component accuracy of 99.5%, by considering the 3 failures described in the previous subsection, and 3D Video has a component accuracy of 99.7% as it introduced 2 additional errors.

A more detailed analysis of the 3D Video errors is shown in Table 5. There have been two failure instances having the root cause originating from the 3D Video itself: a configuration error and a 2D-to-3D handover algorithm error.

3D Video errors have been solved in the following way:*Configuration*: The MQTT client was configured to use a QoS equal to ‘exactly once’ (i.e., QoS = 2) in the 3D Video system. The incidence points out importance of reviewing the implementation, especially any revised one, against the system interface specification formally.*2D-to-3D Alg.*: The 2D-to-3D handover algorithm has been revised to always make an explicit decision about the track direction, including theoretical straight vertical tracks.

The prediction accuracy of 3D Video was further analyzed. L2 error has been utilized in accuracy assessment for location predictions in the 3D video system which is equivalent to that utilized in laboratory test conditions in [27]. The track data captured from the 3D Video system has been used as the best-effort ground truth in calculating L2 error between the actual and predicted data points.

Table 6 summarizes the results which have been computed by averaging track-specific L2 error metrics over all tracks. The average L2-error is 33 cm. Figure 14 shows an overview on distribution of L2 error metrics. Figure 15 presents an example on tracks acquired by the ZED2 sensor and predicted counterparts in the 3D Video system.

### 5.3. Performance of Communication between INSTEAD, MIC and EXTRAS

Success rates of INSTEAD, MIC and EXTRAS communication are presented in Table 7. These have been computed based on the data and logs generated by the INSTEAD components and accounting for checking the compliance of the MIC and EXTRAS message publisher implementations with the architectural specification of INSTEAD.

The rate of MIC to INSTEAD communication success is 93.4%, and it has been computed as the number of MIC positive commuters without failures over all MIC positive commuters. Failures in the data fusion comprise the following scenarios:Attribution of one weapon to two persons: resulting from the association of the same MIC-ID with two 2D Video labels (5 errors)Association of two different MIC-ID detection messages from the same person with two different 2D Video labels (4 errors).Missed MIC detection: resulting from a large time difference between the MIC detection and the 2D Video detection (3 errors).

These failures of associations were mainly due to inaccurate time-synchronization of the machine clocks with the NTP server of the Local Area Network (LAN). This caused that the estimated time difference, based on the MIC and 2D Video detection times, were smaller than the actual time difference of two commuters in some cases. Additionally, the limits of the data fusion algorithm in the capability of combining different MIC detections of the same object, reporting different (estimated) detection times, concurred to the failures. However, as shown by the high number of successes, this was generally not the case.

The rate of INSTEAD to EXTRAS communication success is 97.4%, and it has been computed as the number of triggers received by EXTRAS over the number of commuters. The number of triggers results from the number of correct messages of the INSTEAD handover pipeline except one single message loss in the communication from 3D Video to EXTRAS, due to a congestion in the physical network.

The rate of EXTRAS to INSTEAD communication success is 77.58%, and it has been computed as the number of EXTRAS messages (negative & positive detections), correctly sent to the C&C by EXTRAS, over the number of commuters correctly tracked by INSTEAD before EXTRAS. The lower result is mainly because the EXTRAS detector was integrated in the first days of the trials and interface mismatches between EXTRAS and the C&C had to be solved.

Finally, the overall performance of INSTEAD and the smart glasses (SG) communication was evaluated considering the number of alarms received by the SG over the number of C&C positive detections. The resulting success rate is 68,32%. However, the rate of INSTEAD to SG communication is 99,61% computed as the number of alarms sent by the C&C over the number of positive detections. Considering the period from Day 1 to Day 6 only, the accuracy of the alarm system is of 94,8%. The accuracy much decreased in Day 7 and Day 8, but no alarms confirmations were sent by the SG user in Day 9. The main reason for this is the human factor. In the last days, the glasses were often shared for demonstration purpose and the actual measurement of confirmation messages with the glasses received lower priority.

### 5.4. Computational and Communication Cost

The computational and communication costs are assessed for each of the modules. 2D video processing was performed on one PC with one i7-7820X CPU and two RTX 2080-Ti GPU’s. This was sufficient to process four Full-HD H.264 camera streams at 5 fps for the collection of training material. During the trails, only two 2D cameras were processed without overlapping field-of-view and intermediate cameras were disabled to create a gap between the cameras and to challenge the re-identification algorithm. With some optimization, it was expected that 8–16 cameras could have been processed on one computer. The 3D video processing was performed on an NVIDIA Jetson-Xavier, which allows local edge processing near the 3D sensor without sending the video stream over the network. The main communication cost is the transfer of 2D (and 3D) video data from the cameras to the computers. The 2D video was transferred efficiently with H264 compression from the cameras to the computer. For development and demonstration purpose, the 3D video was also transferred over the network during the trials, but this would not necessary in normal operation due to edge processing near the camera. A critical aspect for the computational cost of the data fusion application might be the in-memory processing capacity of the hosting server to handle the number of concurrent commuters in the monitored environment. For the deployment at the BCT, a PC server with 32 GB memory size has been more than adequate to execute the planned scenarios in real time. The average time duration of a walk by an individual through the monitored area was about 1 min and each run involved groups of varying size, generating about 24 JSON messages over MQTT for a group of 4 people, which could result in 10–200 kbit/s. The JSON messages also contain an URL to retrieve a snippet of a person in case of an alert (Figure 2). Furthermore, the video-sensor message fusion functions are performed in different threads of the C&C to reduce the latency of the processing functions. The average latency in the system appeared to be 243 ms (standard deviation 64 ms), which is based on a measurement over all 586 commuters. The system uses MQTT in Transmission Control Protocol (TCP) mode, which is an OASIS standard lightweight messaging transport where message headers are small to optimize network bandwidth. To guarantee delivery, data re-transmission might take place at peak traffic moments inducing longer communication latencies. The computers are time synchronized using the Network Time Protocol (NTP). Computers running the Linux operating system (2D Video and 3D Video) achieved a synchronization in the scale of about 10ms and on the MS-Windows operating system, an offset of about 10–150 ms was observed.

## 6. Conclusions and Future Work

The INSTEAD fusion engine was developed in the DEXTER program for the management of a plurality of sensors to detect threats without disrupting the flow of commuters. This paper described the results of this fusion engine in a public-space trial in a metro station in Rome. Over multiple days, 586 commuters participated in the trial. The results of the trial showed overall accuracy scores of 97.4% and 97.6% for visualization and hand-over pipelines, respectively, and each component reached high accuracy values (2D Video = 98.0%, Re-identification = 100.0%, 3D Video = 99.7% and C&C = 99.5%). The main causes for errors in the communication with external components were related to unfinished development activities for the sensors, fine-tuning of the software based on the scenario results, and the human factor for the smart glasses.

Advanced privacy-preserving video-based tracking, video-based person re-identification and location prediction were successfully implemented to couple the output of multiple sensors at different locations and to provide security operators with real time information of the critical person with video systems images and position data of each person passing in the monitored area.

The main vulnerability of the system is an incorrect match due to occlusions of people in a crowded environment. Although the re-identification accuracy appeared to be extremely high (no errors of the Re-ID algorithm were observed), there is a chance that people are incorrectly matched, especially if they are occluded in one of the cameras. There are two aspects that should be considered when performing future tests in a crowded environment. The first is that the human oversight should be possible. The human should be able to confirm that the same person was observed near the different sensors. The second aspect is that the handover region should be larger. With the cameras, it was possible to observe every person somewhere in the field of view, even in crowded environments, but due to occlusion it was not always possible to observe every person at the trigger line. Therefore, future work should increase the handover region to improve robustness, assess the performance in an uncontrolled public-space environment and perform a stress-test for high-density crowds.

## Figures and Tables

**Figure 1 sensors-23-00440-f001:**
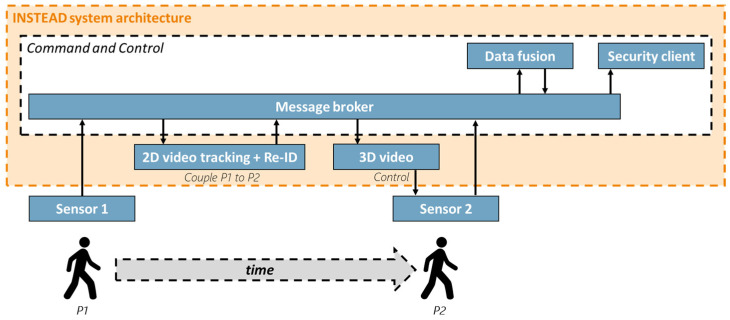
INSTEAD system architecture.

**Figure 2 sensors-23-00440-f002:**
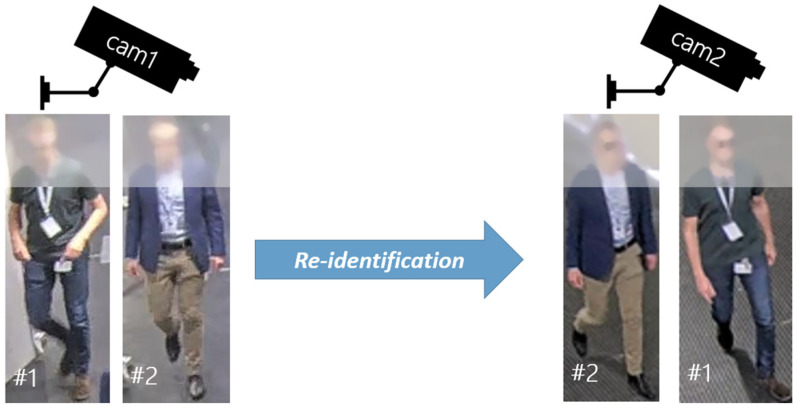
Person re-identification with anonymized faces.

**Figure 3 sensors-23-00440-f003:**
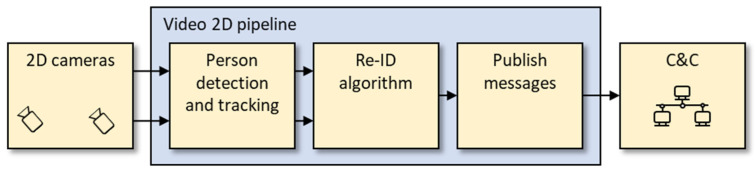
Architecture of 2D Video.

**Figure 4 sensors-23-00440-f004:**
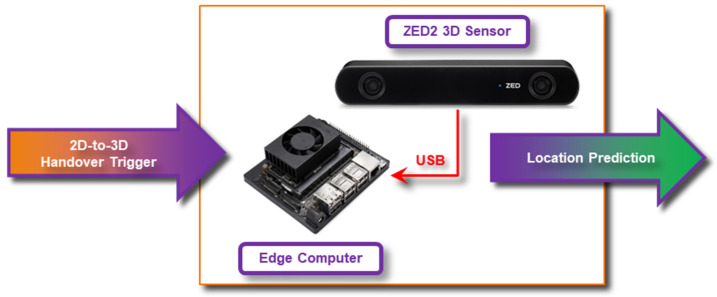
3D Video system for predictions.

**Figure 5 sensors-23-00440-f005:**
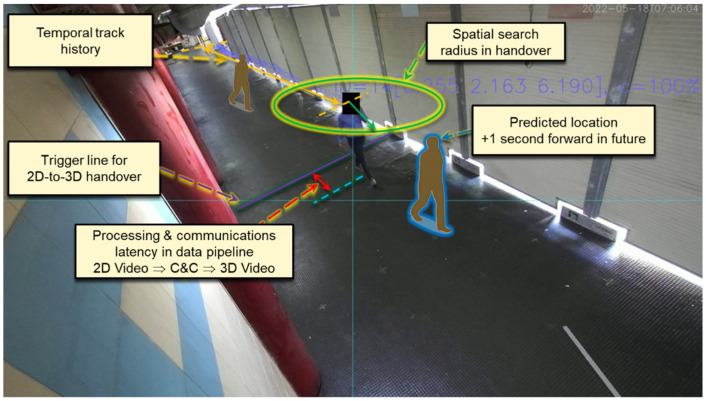
2D-to-3D Video system handover.

**Figure 6 sensors-23-00440-f006:**
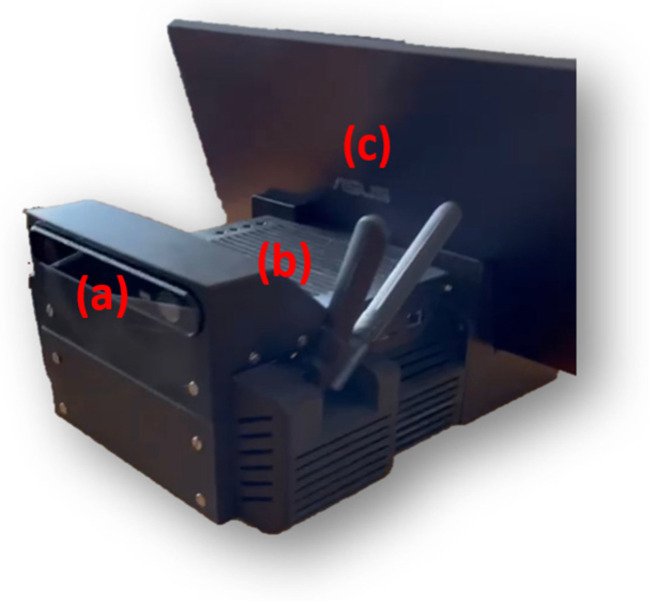
3DSentinel sensor. (**a**) Stereolab 3DSentinelZED camera; (**b**) NVIDIA Xavier PC; (**c**) Removable Monitor for setting sensor parameters.

**Figure 7 sensors-23-00440-f007:**
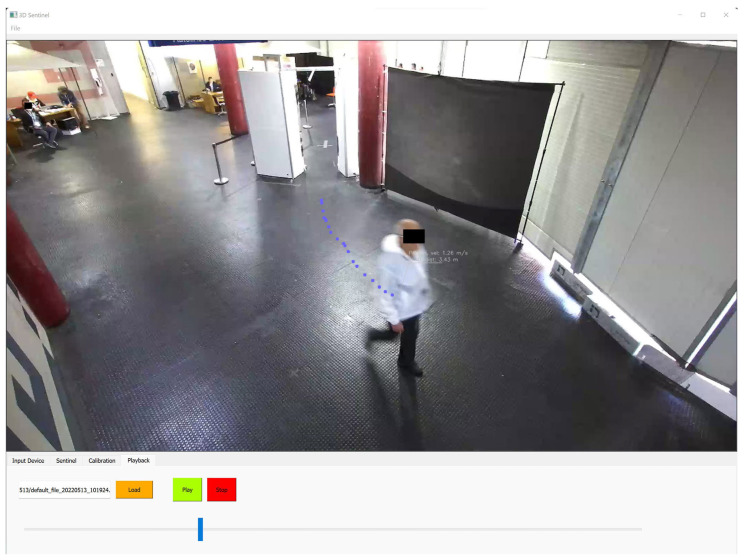
3DSentinel software interface: the blue dots are the trace made by the detected person.

**Figure 8 sensors-23-00440-f008:**
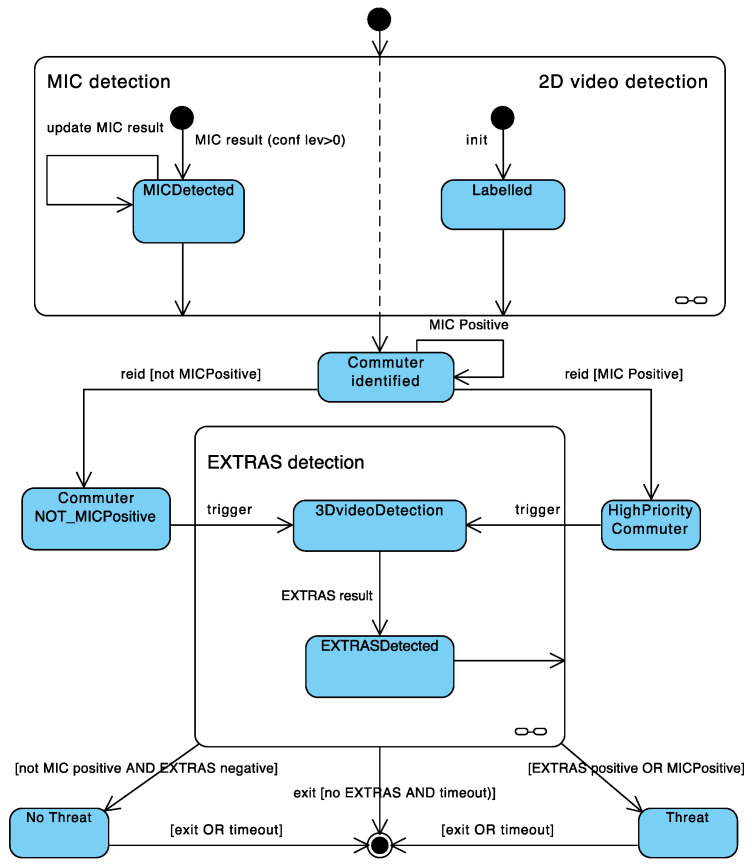
State-chart specification of the data fusion method.

**Figure 9 sensors-23-00440-f009:**
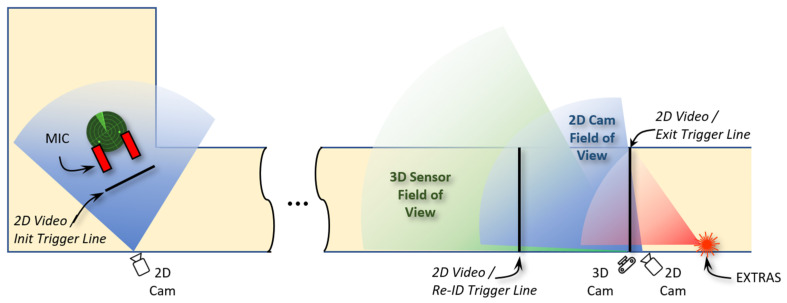
Map of the environment at the Anagnina metro station.

**Figure 10 sensors-23-00440-f010:**
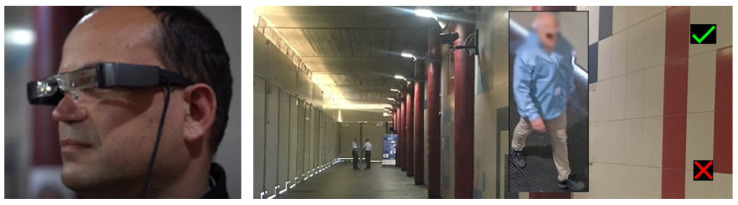
Detections are visualized on smart glasses (**left**). The person related to the weapon or explosive detection is displayed as an overlay on the smart glasses (**right**).

**Figure 11 sensors-23-00440-f011:**
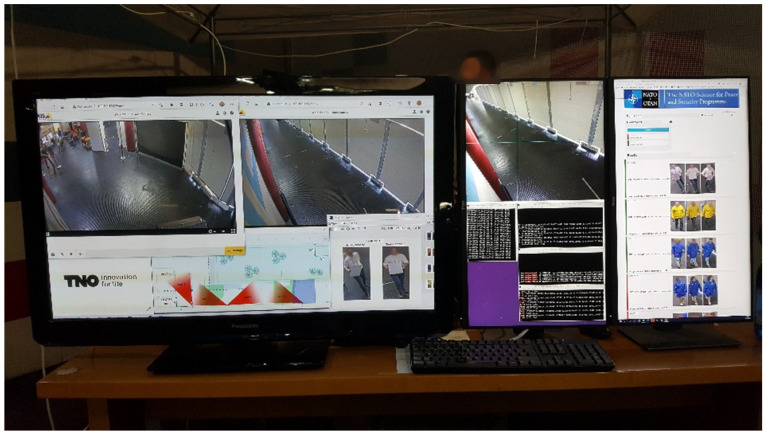
Graphical user interfaces of 2D-video (**left**), 3D-video (**middle**) and C&C (**right**).

**Figure 12 sensors-23-00440-f012:**
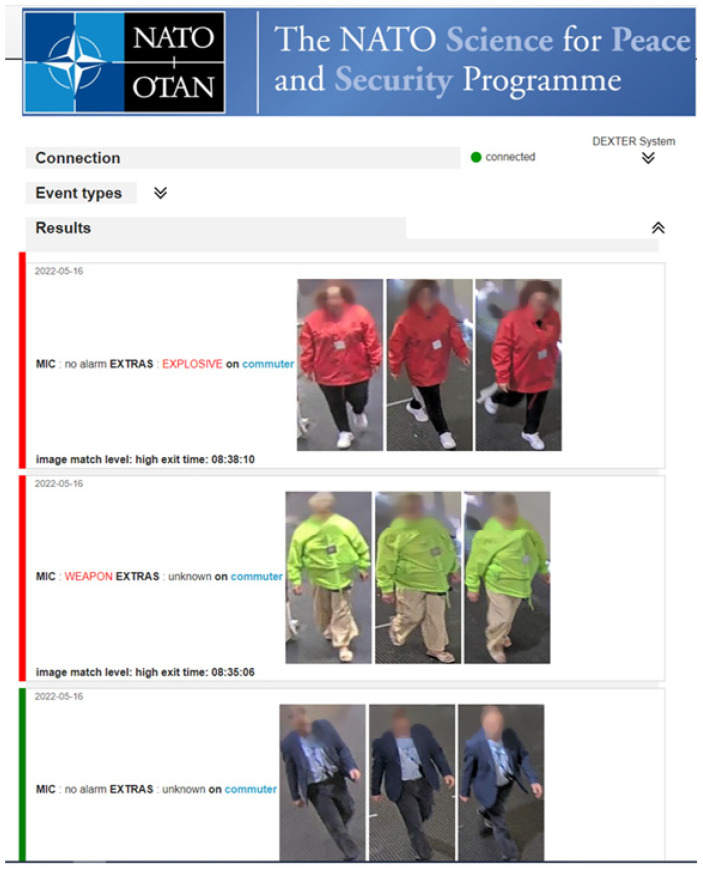
Graphical user interface of C&C, based on [38], with explosive detection (**top**), weapon detection (**middle**) and no detection (**bottom**).

**Figure 13 sensors-23-00440-f013:**
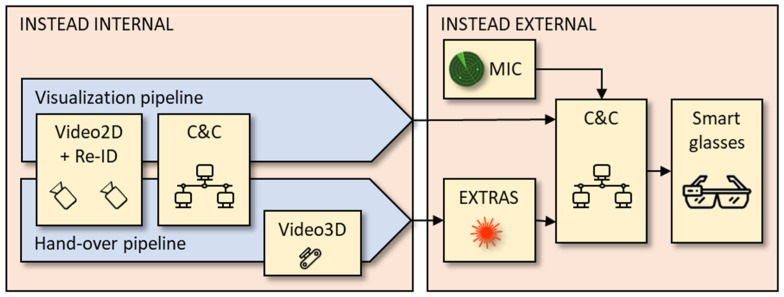
Two INSTEAD pipelines: visualization and hand-over.

**Figure 14 sensors-23-00440-f014:**
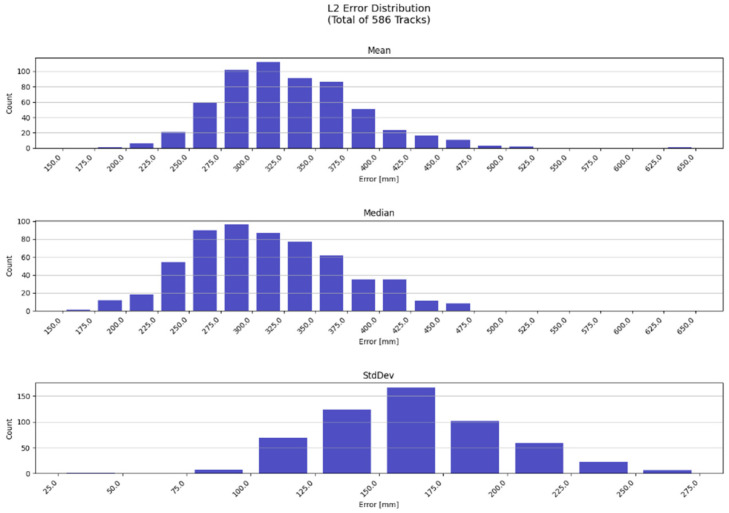
L2-error distribution for predicted tracks in test runs in 3D Video.

**Figure 15 sensors-23-00440-f015:**
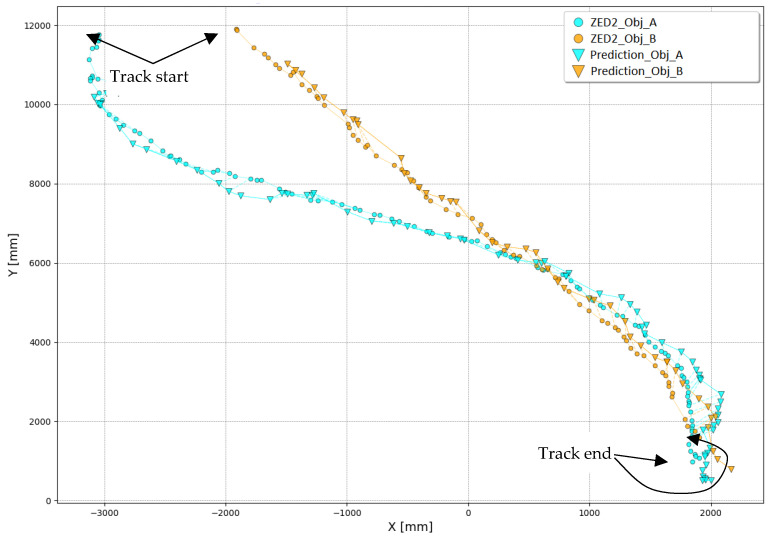
Two examples on raw 3D Video data and computed predictions.

**Table 1 sensors-23-00440-t001:** Number of runs and number of commuters.

Day ID	1	2	3	4	5	6	7	8	9	Total
Total runs	20	25	32	53	53	44	24	20	16	287
Total commuters	29	37	47	76	80	95	96	74	52	586

**Table 2 sensors-23-00440-t002:** Performance of the visualization pipeline.

Component	Total (#)	Overall Accuracy (%)	Component Accuracy (%)
Commuters	586	-	-
2D Video correct messages	574	98.0	98.0
C&C correct messages	571	97.4	99.5

**Table 3 sensors-23-00440-t003:** Error causes of 2D Video.

Label	Freq.	Cause
Detection	7	Subjects were too close behind each other near the crossline near MIC (the time interval was less than 2.0 s), causing a missing image due to occlusion.
Reappear	2	Same person entered the region twice within a short time frame (within 4.0 min). The second appearance matched against first.
Re-ID alg.	0	No errors due to the re-identification algorithm itself.
Publish	2	An incorrect person-ID or image URL was published by the 2D Video. Internal logs showed a correct match of the system.
C&C	1	C&C displayed the images with large delay, which is possibly related to late response of other modules. Correct messages were sent in a timely manner by 2D Video.
**TOTAL**	**12**	

**Table 4 sensors-23-00440-t004:** Performance of the hand-over pipeline.

Component	Total (#)	Overall Accuracy (%)	Component Accuracy (%)
Commuters	586	-	-
2D Video correct messages	577	98.5	98.5
C&C correct messages	574	98.0	99.5
3D Video correct messages	572	97.6	99.7

**Table 5 sensors-23-00440-t005:** Error causes of 3D Video.

Label	Freq.	Cause
Configuration	1	An MQTT message lost due to a configuration error in the MQTT client in the 3D Video system; the configured Quality of Service has been equal to ‘at most once’ while it should have been QoS equal to ‘exactly once’.
2D-to-3D Alg.	1	Available 3D track data has had a very vertical trajectory in data points (i.e., parallel along the Y-axis) at the 2D-to-3D handover time point, and 2D-to-3D handover algorithm has been inconclusive what is the track direction (forward vs. backward).
**TOTAL**	**2**	

**Table 6 sensors-23-00440-t006:** L2 error metrics in 3D video system.

Metric	L2 Error (mm)
Mean	329
Median	313
Standard deviation	173

**Table 7 sensors-23-00440-t007:** Results of INSTEAD, MIC and EXTRAS communication.

Communication	Accuracy
MIC-INSTEAD	93.4%
INSTEAD-EXTRAS	97.4%
EXTRAS-INSTEAD	77.6%

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
