# Peer review of "An Integrated Fusion Engine for Early Threat Detection Demonstrated in Public-Space Trials"

_sensors, 2022, doi:10.3390/s23010440_

Round 1

Reviewer 2 Report

- The introduction section should be enhanced by discussing more about the current solutions and their limitations that motivated this study.

- Discuss in detail about the key contributions of this work.

- A section, recent works can be added to summarize the current SOA.

- Some of the recent works on digital forensics such as the following can be discussed

A comprehensive survey on digital video forensics: Taxonomy, challenges, and future directions

- The author should enhance the results section by comparing the results obtained in this study with recent state of the art.

- What are the computational and communication cost of the proposed framework?

- What are the threats to validity of the proposed methodology?

- The authors should also discuss about the possible future directions that will help the researchers to further work in this domain.

-

Round 2

Reviewer 1 Report

The author has made modifications as required, and the current version meets the requirements.

Reviewer 2 Report

All the comments are addressed by the authors.